# Revealing the True Morphological Structure of Macroporous Soft Hydrogels for Tissue Engineering

**Bohumila Podhorská**[ID]**, Miroslav Vetrík, Eva Chylíková-Krumbholcová, Lucie Kománková, Niloufar Rashedi Banafshehvaragh, Miroslav Šlouf**[ID]**, Miroslava Dušková-Smrčková**[ID] **and Olga Janoušková \***[ID]

Institute of Macromolecular Chemistry of the Czech Academy of Sciences, Heyrovského náměstí 2, 162 06 Prague 6, Czech Republic; podhorska@imc.cas.cz (B.P.); vetrix@seznam.cz (M.V.); chylikova@imc.cas.cz (E.C.-K.); luciek2@seznam.cz (L.K.); rashedi@imc.cas.cz (N.R.B.); slouf@imc.cas.cz (M.Š.); m.duskova@imc.cas.cz (M.D.-S.)

**\*** Correspondence: janouskova@imc.cas.cz



**Featured Application: The findings in our study can significantly help in choosing conditions for 3D hydrogel characterization for application in tissue engineering.**

**Abstract:** (1) Background: Macroporous hydrogel scaffolds based on poly [N-(2-hydroxypropyl) methacrylamide] are one of the widely studied biocompatible materials for tissue reparation and regeneration. This study investigated the morphological changes during hydrogel characterization which can significantly influence their future application. (2) Methods: Three types of macroporous soft hydrogels differing in pore size were prepared. The macroporosity was achieved by the addition of sacrificial template particles of sodium chloride of various sizes (0–30, 30–50, and 50–90 μm) to the polymerizing mixture. The 3D structure of the hydrogels was then investigated by scanning electron microscopy (SEM) and laser scanning confocal microscopy (LSCM). The SEM was performed with specimens rapidly frozen to various temperatures, while non-frozen gels were visualized with LSCM. (3 and 4) Results and Conclusion: In comparison to LSCM, the SEM images revealed a significant alteration in the mean pore size and appearance of newly formed multiple connections between the pores, depending on the freezing conditions. Additionally, after freezing for SEM, the gel matrix between the pores and the fine pores collapsed. LSCM visualization aided the understanding of the dynamics of pore generation using sodium chloride, providing the direct observation of hydrogel scaffolds with the growing cells. Moreover, the reconstructed confocal z-stacks were a promising tool to quantify the swollen hydrogel volume reconstruction which is not possible with SEM.

**Keywords:** hydrogel scaffolds; laser scanning confocal microscopy; scanning electron microscopy; pHPMA; cell cultivation

## 1. Introduction

Synthetic macroporous hydrogel scaffolds have been successfully applied as temporal and permanent templates for cell cultivation and tissue regeneration purposes due to their good biocompatibility and biomimetic features, such as highwater retention, tailorable microstructure, tissue-like stiffness, and rapid nutrient diffusion [1–3]. They also can be tuned to promote cell growth, migration, or angiogenesis [4,5]. Hydrogel scaffolds must provide suitable mechanical properties as well as porous structure morphology and connectivity to enable cell growth [2,6]. The pore connectivity, size, and shape are some of the major factors for appropriate cell ingrowth [7], with the optimal pore size differing according to the type of tissue—e.g., 5 μm for neovascularization [8], 5–15 μm

for fibroblast ingrowth [9], 20 μm for hepatocyte ingrowth [10], or 20–125 μm for adult mammalian skin [11].

The characteristics of pore properties, such as their dimension, distribution, volume, interconnectivity, shape roughness between pores, and overall pore shape, must be carefully tailored during scaffold preparation. These properties are crucial for new tissue growth and reorganization, which include building the cellular network and interconnected pathways for nutrient transportation, cell signaling, and proliferation. The porosity of a hydrogel can be achieved by various techniques such as salt/porogen templating, including oriented pore fabrication [12,13], bi-continuous emulsion templating [14], gas foaming [15], cryogelation [16,17], 3D printing, or electrospinning [18], which all can provide a different level of control of the pore size, distribution, and connectivity.

The porogen leaching technique is a well-established effective method of introducing pores of the desired shape, size, and orientation to the gel matrix. With the advantage of ease of use and biological safeness, different materials such as sodium chloride [12], ammonium oxalate crystals [19], and paraffin [20] have been used. However, there are challenges related to appropriate design or porogen packing to achieve connected porosity on one side and a coherent gel matrix on the other side through tuned concentration and the placement of particles in the polymerization mixture [21,22].

The most frequently used technique to determine the morphology of pores in hydrogels is based on scanning electron microscopy (SEM) [23,24]. These methods offer a high resolution and large fields of the depth of the samples. Conventional SEM cannot adopt wet samples in the electron beam chamber because vaporized water molecules would interfere with the primary or secondary electron beam. Thus, the reduction in the vapor pressure of possible evaporating liquids, such as water, from samples, is a typical condition for electron detection. Swollen hydrogel samples characterized by the SEM method must be freeze-dried and typically metal-sputtered to enhance the surface reflectivity depending on the SEM method. The drying process of soft hydrogel samples and the solidification of swollen gel (swelling and polymerizing in casting resin) can substantially change the morphology, leading to secondary porosity formation. The freezing process itself greatly differs depending on factors such as the freezing media, freezing rate, or size of the freezing sample. The effect of freezing on the internal structure of highly swollen hydrogels and the possibility of the formation of secondary pores due to ice crystal formation and growth during freezing was previously described by Aston and colleagues for alginates [25], and by Paterson and colleagues for poly (2-hydroxyethyl methacrylate), pHEMA [26]. Basically, water present in the hydrogel freezes and crystalizes within the gel matrix; the softer the gel matrix, the more severe the deformation. The process of water removal from swollen hydrogels by conventional drying causes large changes in the total volume, typically involving a transition of the polymeric matrix to glassy state, which can be accompanied by mechanical failure, the deformation of pore shape, the shrinking of the gel wall, and the collapse of the small pores [27]. However, some attempts to resolve these problems have been addressed using cryogenic microscopic methods. The idea is to rapidly quench samples in order to avoid water crystallization and study samples in native conditions. The faster quenching of water leads to an amorphous glass, impeding the nucleation of crystals. A recent study by Parmenter et al. (2016) showed the possibility of using cryogenic focused ion beam scanning electron microscopy to visual the morphology while preserving the water volume content in samples, thereby avoiding structure collapse by shrinking, also involving gel plunge freezing in the preparatory process [28]. Another approach avoiding sample freezing is used in environmental scanning electron microscopy (ESEM) [29]. Nevertheless, this method has still some drawbacks compared to SEM—e.g., its lower resolution or decrease in contrast.

Indeed, studying the gel in its native environment, the equilibrium swollen state is the most optimal option and is possible with laser scanning confocal microscopy (LSCM), which allows the visualization of swollen hydrogels in both 2D and 3D. LSCM allows the direct observation of samples in their hydrated state without the necessity of freezing, collecting z-plane images (z-stacks) with subsequent gel volume reconstruction to provide information about the size and distribution of pores

in hydrogels [30]. Moreover, LSCM enables the observation of native hydrogels, as well as hydrogels with growing cells in situ [31,32].

This work focused on the detailed assessment of the morphology of soft porous non-degradable synthetic hydrogels based on covalently crosslinked poly [N-(2-hydroxypropyl) methacrylamide] (pHPMA). Porous structures were created using solid washable porogen particles, which served as pore templates that were added during gel synthesis. Morphological images were obtained by SEM and LSCM methods and compared to assess the relationship of the SEM images to the morphology of the swollen gel state. Furthermore, we determined whether the material images obtained by SEM agree with the LSCM observations and can provide relevant information regarding pore size and connectivity in systems where the equilibrium swollen state plays a major role. Prior to SEM analysis, pHPMA hydrogels were freeze-dried under different freezing conditions and the effects on the morphological porous structure were examined. LSCM scans of equilibrium swollen hydrogels were used to reconstruct the 3D gel volume structure using an image analysis processing package, allowing the visualization of the cell growth of hydrogels in situ.

## 2. Materials and Methods

### 2.1. Materials

Ethylene glycol dimethacrylate (EDMA), 2,2'-azobisisobutyronitrile (AIBN), polyethylene glycol 400 (PEG 400), N-(3-Aminopropyl) methacrylamide hydrochloride, [2-(methacryloyloxy)ethyl] trimethylammonium chloride (MOETACl) isolated from water solution, trypsin, L-ornithine, and laminin were purchased from Merck(Darmstadt, Germany). Dulbecco's modified Eagle's medium (DMEM) was purchased from ThermoFischer Scientific (Brno, Czech Republic). Penicillin, streptomycin, and Hoechst were purchased from ThermoFischer Scientific (Brno, Czech Republic). Sodium chloride cryst., p.a., was purchased from Lach-ner, s.r.o. (Neratovice, Czech Republic). N-(2-hydroxypropyl) methacrylamide (HPMA) was synthetized according to a method described earlier [33].

### 2.2. Preparation of Porogen Particles

Sodium chloride crystals (250 g) were milled using the laboratory ball mill Fritsch Analysette 3 Spartan overnight. The salt powder was fractionated by sieving overnight using a set of Retsch stainless steel sieves of pore sizes 90, 50, and 30 μm, respectively (according to standard DIN-ISO 3310/1—which specifies the technical requirements and corresponding test methods for test sieves of metal wire cloth.).

### 2.3. Preparation of Hydrogels

Hydrogel scaffolds were prepared by thermally initiated free radical polymerization. The monomeric mixture of HPMA, 2-(methacryloyloxy)ethyl]trimethylammonium chloride (MOETACl), methacrylated fluorescein, and ethylene glycol dimethacrylate (EDMA) crosslinker were dissolved in PEG 400 (21.25 wt%) in the molar ratio 96.3:1.7:1:1. The radical initiator 2,2'-azobisisobutyronitrile (AIBN) (2.6 wt% in monomers) was added to the mixture and dissolved, then sodium chloride particles were quickly added to the polymerization mixture (87.5 wt%) and mixed well. The formed paste was homogenized by mixing; transferred into the polymerization mold, as described previously [12]; and heated in the oven at 80 °C for 16 h. The prepared pellets were removed and placed in an excess of water (400 mL) at 37 °C. The samples were slowly shaken and kept in the water bath for two weeks, with the water replaced approximately every 24 h. Hydrogels were prepared using three sizes of salt porogen particles: 1–30 μm (group 1), 30–50 μm (group 2), and 50–90 μm (group 3). The reference hydrogel without salt particles, termed the basic hydrogel matrix, was prepared from the same polymerization mixture and under the same polymerization conditions.

### 2.4. Methacryloylation of Fluorescein

Fluorescein 5-isothiocyanate (0.5 g, 1.3 mmol) was dissolved in 100 mL of anhydrous tetrahydrofuran (THF) following the addition of triethylamine (0.4 mL, 3 mmol), then the solution was cooled down to 0 °C. N-(3-aminopropyl)methacrylamide hydrochloride (0.12 g, 1.5 mmol) was diluted in 5 ml of THF and added dropwise to the chilled mixture. The reaction occurs at room temperature for 12 h. Then, the solution was evaporated under a vacuum and purified by column chromatography using the mobile phase CHCl3: acetone (9:1), with the reaction yield in an orange solid product (0.48 g).

### 2.5. Scanning Electron Microscopy (SEM)

The hydrogel morphology was studied using an electron microscope TESCAN Vega Plus TS 5135 (Tescan Brno, Czech Republic) in a high vacuum and secondary electron imaging mode. Three samples from each group were separately immersed in distilled water (10 mL) for several days. After they reached swelling equilibrium, the samples were cut into sections (10 × 7 × 4 mm) and frozen either in a freezer at −23 °C, dry ice at −78 °C, or liquid nitrogen at −195 °C. All the frozen samples were then lyophilized for three days (at vacuum 0.1 mbar), cut into thin slices (1.5 mm), and covered with a 4 nm-layer of platinum using a sputter coater LEICA EM SCD050 (Leica Microsystems). Images of hydrogel samples with different porosities and prepared by different freezing methods were analyzed using the ImageJ software with respect to certain constraints.

### 2.6. Laser Scanning Confocal Microscopy (LSCM)

Hydrogels were visualized using the Olympus IX83 multi-photon LSCM FV10-ASW and scanned using the 10× objective Plan ApoN (1.42 numerical apertures). The ImageJ software was utilized to determine the average pore size of the hydrogels, the pore distribution, and the reconstruction of the hydrogel 3D structures.

### 2.7. Three-Dimensional Computer Reconstructions

Sets of z-planes of swollen hydrogels differing in pore size were obtained by LSCM (c.f. Part 2.6.) using the Olympus FLUOVIEW FV1000 software. Imported sets of LSCM images included 202, 216, and 386 TIF files (the thicknesses of the analyzed gel sections were 30, 50, and 90 μm). The distance between the scanned plane slices was 0.25 μm, and the scanned area measured 400 × 400 μm. The 3D computer reconstruction from the acquired z-stacks was processed using the Imaris software, with the different phases (gel vs. pores filled with liquid) distinguished by adjusting the pixel color intensity threshold value to determine the gel borders. Afterwards, the "region grow" operation connected all the gel areas together, then, similarly, the pore area was identified using the "cavity fill" operation. The counts of voxels in the gel matrix and in the pores were used to estimate the pore volume fraction detectable from the images, with $\phi_{CT}$ as the ratio of the number of voxels in pores divided by the total number of voxels.

### 2.8. Swelling of Hydrogels

The equilibrium swelling of hydrogels in water at laboratory temperature (25 °C) was determined gravimetrically. The water content in swollen gels was expressed as the weight percentage of water in the swollen system hydrogel-water according to Equation (1):

$$SD = 100 \times (m_{sw} - m_d)/m_{sw}, \tag{1}$$

where $m_{sw}$ is the weight of swollen hydrogel in equilibrium (i.e., after the extraction of PEG400, salt template particles, and a possible unreacted portion of reactive systems-sol), and $m_d$ is the weight of dry gels obtained after swelling by drying the samples to a constant weight in an oven under reduced pressure to 5 mbar at 70 °C. For each system including the basic hydrogel matrix without the

pores from salt templating, the swelling test was performed with three samples of gel and the average value was calculated.

*2.9. Gravimetric Determination of Pore Volume in Swollen Gel*

The volume fraction of pores in the swollen gel, $\phi_{por}$, was determined using the equilibrium swelling of the macroporous hydrogels and the swelling of the reference hydrogel matrix prepared without the pores $sd_{por}$ and $sd_{mx}$, respectively. This reasoning of volume balance was adopted assuming that the swelling of gels was isotropic. The volume balance in the swollen system is as follows:

$$V_{sw} = V_{PL} + V_{mx}, \tag{2}$$

where $V_{sw}$ is the total volume of swollen gel; $V_{PL}$ is the "empty" volume of pores after the salt particles were washed and at swelling equilibrium, attained after the extraction process described above; and $V_{mx}$ is the volume of the swollen basic gel matrix, determined experimentally with a specially prepared reference sample without salt particles and calculated using the following equation:

$$V_{mx} = (m_d/\rho_{HPMA} + m_{wm}/\rho_w), \tag{3}$$

where $m_{wm}$ is the weight of water in grams swollen per dry gel of weight $m_d$ in grams for the reference (non-porous) hydrogel matrix, $\rho_{HPMA} = 1.14$ g·cm$^{-3}$ is the specific gravity of the dry hydrogel matrix adopted from ref. [34], and $\rho_w$ is the specific gravity of water at 25 °C (all rhos given in g·cm$^{-3}$). The $V_{mx}$ then provides the volume of the swollen matrix gel produced per 1 g of dry matrix. The volume of pores formed after the salt particles were washed, $V_{PL}$, can be then expressed in the cm$^3$ per g of dry macroporous gel using the weights measured in the swelling experiment and the specific densities given above:

$$V_{PL} = V_{SW} - V_{mx} = (m_d/\rho_{HPMA} + m_w/\rho_w) - (m_d/\rho_{HPMA} + m_{wm}/\rho_w), \tag{4}$$

where $m_w$ is the total weight of water in the swollen gel (g), including the water in pores formed after the salt particles and water were swollen in the matrix, $m_d$ (g). The other symbols were described above in Equations (1)–(3). Now, the swelling degree values for macroporous gels, $sd_{por}$, and the reference matrix gel, $sd_{mx}$, are defined as:

$$sd_{por} = (m_{sw} - m_d)/m_{sw}, \, sd_{mx} = (m_{swm} - m_d)/m_{swm}. \tag{5}$$

Equation (4) provides the volume of pores in cm$^3$ produced by 1 g of dry macroporous hydrogel (cm$^3$/g dry gel), and can be re-written as:

$$V_{PL} = (1/\rho_{HPMA} + sd_{por}/((1 - sd_{por})\rho_{aq})) - (1/\rho_{HPMA} + sd_{mx}/((1 - sd_{mx})\rho_{aq})). \tag{6}$$

The total swollen volume in cm$_3$ obtained by the swelling of 1 g of dry macroporous gel is:

$$V_{SW} = (1/\rho_{HPMA} + sd_{por}/((1 - sd_{por})). \tag{7}$$

The pore volume fraction, $\phi_{por}$, is then given as:

$$\Phi_{por} = V_{PL}/V_{SW}. \tag{8}$$

*2.10. Cell Growth on Hydrogel Scaffolds*

The rat mesenchymal stem cells (rMSCs) were kindly provided by Dr. P. Jendelova, Institute of Experimental Medicine of the Czech Academy of Sciences, and were cultivated in DMEM supplemented with fetal bovine serum (FBS), 100 units of penicillin, and 100 µg·mL$^{-1}$ streptomycin in 25 cm$^2$ flasks

in a humidified incubator at 37 °C with 5% $CO_2$. Cells ($1 \times 10^5$ cells per 1 mL) were added to the wells of 24-well flat-bottom plates with a $4 \times 4 \times 4$ mm gel section. Prior to cell-seeding, the hydrogel samples were sterilized by UV and soaked overnight in poly-l-ornithine solution (0.01%) diluted 1:6 in distilled water at 37 °C. The gels were then washed in deionized water and incubated in a laminin solution (laminin diluted in DMEM) at a final concentration of 10 µg·mL$^{-1}$ of laminin at 37 °C for 2 h. The seeding tests for each type of scaffold were performed in duplicate.

After three or five days of cell cultivation on pHPMA hydrogels, the cells were washed three times with PBS and the cell growth was observed via LSCM. The cell nuclei were stained with Hoechst (5 µg·mL$^{-1}$) for 10 min before imaging, and the number of viable cells growing in yjr hydrogels was evaluated using the Alamar Blue cell viability assay (ThermoFischer Scientific, Brno, Czech Republic), as described previously [32]. Briefly, a section of hydrogel was added to an empty well in 300 µL of fresh media, then 30 µL of AlamarBlue cell viability reagent containing resazurin, a compound which is reduced to fluorescently active resorufin in metabolically active cells, was added to the media with hydrogels. After 4 h of incubation, the fluorescence was measured using a multi-well plate reader SynergyNeo (Bio-Tek, iBiotech, Prague, Czech Republic), with excitation at 570 nm and emission at 600 nm. The fluorescence intensity directly correlates with the number of viable cells, and the absolute cell number was calculated from the calibration curve.

## 3. Results and Discussion

### 3.1. Porous pHPMA Hydrogel Preparation and Morphology

Porous hydrogel scaffolds were prepared by the thermally initiated free radical polymerization of main monomer HPMA, copolymerized with co-monomer MOETACl and methacryloylated fluorescein and crosslinked with crosslinker EDMA, and the morphological features were evaluated using SEM and LSCM (Figure 1). The equilibrium swollen hydrogels examined by the SEM analysis were frozen using three different common freezing temperatures: (a) −23 °C (freezer), (b) −78 °C (dry ice), and (c) −195 °C (liquid nitrogen) (Figure 1a–c). Then, the gels were freeze-dried, sputtered with Pt, and analyzed by SEM. The hydrogel samples evaluated by LSCM were visualized directly in the equilibrium swollen state without further sample processing (Figure 1d) and were not treated by freezing or temperature change. The notation of the freezing temperatures and hydrogel types coded according to their porogen size range is shown in Table 1.

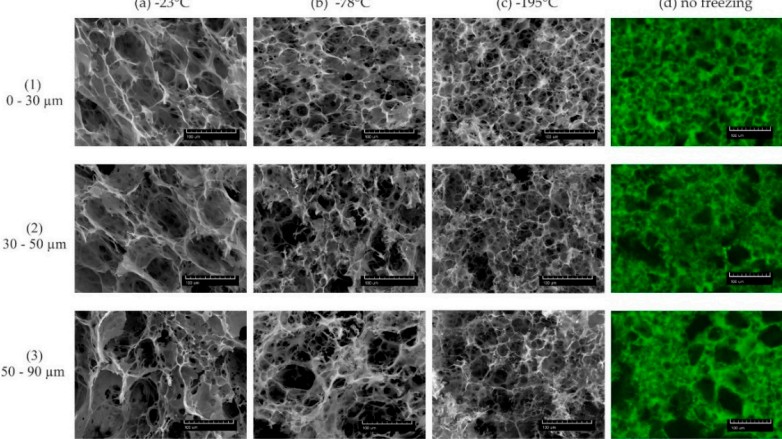

**Figure 1.** Comparison of the pHPMA hydrogel morphology visualized by SEM (**a**–**c**) and LSCM (**d**). Set 1 (row 1) was prepared using porogen 0–30 µm, set 2 (row 2) using porogen 30–50 µm, and set 3 (row 3) using porogen 50–90 µm, with freezing conditions as follows: group (**a**) −23 °C, group (**b**) −78 °C, group (**c**) −195 °C. LSCM images of pHMPA hydrogel represent a group (**d**) scanned in the equilibrium swollen state of gels.

**Table 1.** Notation of the pHPMA hydrogel samples according to the sample preparation conditions for the visualization of the hydrogel morphology by SEM and LSCM.

| Groups/Sets | (a) −23 °C | (b) −78 °C | (c) −195 °C | (d) No Freezing |
|---|---|---|---|---|
| **1** (0–30 μm) | 1a | 1b | 1c | 1d |
| **2** (30–50 μm) | 2a | 2b | 2c | 2d |
| **3** (50–90 μm) | 3a | 3b | 3c | 3d |

Each set of pore sizes in Figure 1 (series in rows) was prepared using the fractionated porogen of the same size range. The visual evaluation of the images in Figure 1 shows that different freezing temperatures in the case of SEM can cause remarkable changes in the porous morphology and pore size. The samples prepared for SEM analysis using the same freezing method are more similar than the different porogen size samples (Figure 1 columns vs. rows). The changes in the morphology of the pores are the most visible in the samples frozen at the lowest freezing temperature and lessened with a decreasing temperature. The comparison of the morphologies depicted in Figure 1d, where LCSM was used for the visualization of hydrogels, indicates that the freezing of gels can influence the soft hydrogel morphology during the sample preparation for SEM analysis. A small amount of incorporated fluorescent probe that was added as a co-monomer during the hydrogel network formation allows the non-invasive visualization of the sample by the LSCM technique, with the respect to the wall thickness, pore size distribution, and shape. Moreover, hydrogel morphology examination by LSCM can be conducted in different solutions, such as PBS or biological medium, respecting the swelling properties in different aqueous media. Based on these factors, we propose that the depicted morphologies in Figure 1d accurately describe the primary hydrogel microstructure in its native equilibrium swollen state.

The effect of freezing temperatures on the internal structure of highly swollen hydrogels was previously reported [25,26,35]. In line with these observations, the present study suggests that the formation of ice crystals during the freeze-drying process could modify the resulting hydrogel morphological structure. Moreover, the secondary pores formed due to ice crystal formation and growth during freezing can dramatically change the morphology of pore walls. Our observations suggest how the preparation of the soft microporous hydrogel sample for morphological characterization can influence the hydrogel morphology, as the porosity is due to the formation of ice crystals. Based on these observations, we propose that the interpretation of SEM images during the morphological examination can differ depending on the sample preparation.

The direct measurements of the pore sizes in all samples in Figure 1 (SEM and LSCM, 400 × 400 μm) were quantified using the ImageJ software, as described previously [32]. For each gel type, three different microphotographs were evaluated (see Figure S1). The values averaged as shown in Table 2.

**Table 2.** Evaluation of the pore sizes in hydrogels prepared under different freezing conditions and with various porogen fractions. Minimum and maximum pore sizes found in SEM (a–c)) and LSCM (d) micrographs.

| Pore Size | 1 (0–30 μm) Min | Max | 2 (30–50 μm) Min | Max | 3 (50–90 μm) Min | Max |
|---|---|---|---|---|---|---|
| **(a) −23 °C** | 8 | 76 | 6 | 107 | 11 | 134 |
| **(b) −78 °C** | 8 | 48 | 9 | 68 | 6 | 97 |
| **(c) −195 °C** | 9 | 41 | 10 | 70 | 6 | 75 |
| **(d) No Freezing** | 8 | 35 | 8 | 53 | 8 | 90 |

The analysis of the pore size distribution by the ImageJ software shows the increasing tendency of the pore size distribution with increasing porogen size for samples prepared with 0–30 and

30–50 μm-sized particles. Interestingly, as the size of the porogen increased, the span of the pores also increased, possibly due to the presence of small porogen particle "fine dust", as described in the section: salt porogen features.

The largest pores were generated when swollen hydrogels were frozen at −23 °C. With the lowering of the freezing temperature, the pore size decreased—c.f., groups (b) and (c) in Figure 1. This effect corresponds to the freezing rate; when a faster freezing temperature rate is applied, smaller ice crystals are formed [25,36].

The comparison of the morphological structure gained by LSCM revealed interesting structural features that were hidden when the SEM method was used—e.g., the relation of the shape of the pores with the shape of the porogen crystals embedded in the matrix during polymerization (Figure 2). In Figure 2c, it is clear that under freezing conditions, the gel phase collapsed to thin shells or fiber-like formations and the pores lost their resemblance to imprints after the salt particles (Figure 2b). This further confirms that the complex evaluation of soft hydrogel porous morphology and the quantification of porometric features (pore volume, pore connectivity, minimum pore-channel size, the volume of connected pores, the volume of separated pores) are more accurate based on data from LSCM.

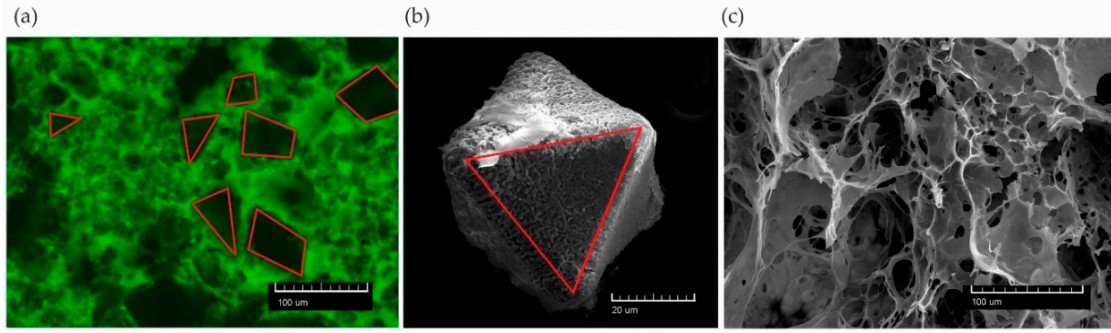

**Figure 2.** Comparison of the structure of sample set 3 (50–90 μm) image created by LSCM performed with unfrozen gel (**a**) and the SEM of gel previously frozen to −195 °C (**c**) with salt crystal as porogen (**b**). Selected single pores are highlighted in red.

### 3.2. Salt Porogen Features

The measurement of the distribution of pore sizes after the salt particles were washed shows a higher abundance of smaller pores than the porogen fractionation when a lower sieve size was used. These fine pores below the "sieving limit" were remarkable, especially in hydrogels prepared with a porogen in the size ranges of 30–50 and 50–90 μm, as indicated in Table 2 and Figure 1d. The distribution of the sizes of solid porogen evaluated by SEM shows that the fractionated salt contained mostly irregular particles with an aspect ratio close to 1. Besides larger particles that fit the sieving ranges, all the porogen fractions contained some amount of very fine particles of sizes approximately below 10 μm held in larger clusters. These clusters were probably formed during sieving (or even existed before), but could be dispersed in the polymerization mixture during the gel preparation. The formation of clusters was most likely caused by environmental humidity, allowing the salt particles to connect by moisture, perhaps with some contribution of electrostatic effects. Thus, in each porogen size fraction, a certain amount of a "fine dust" was present, causing in the hydrogel the formation of pores below the sieve size range; see Figure 3.

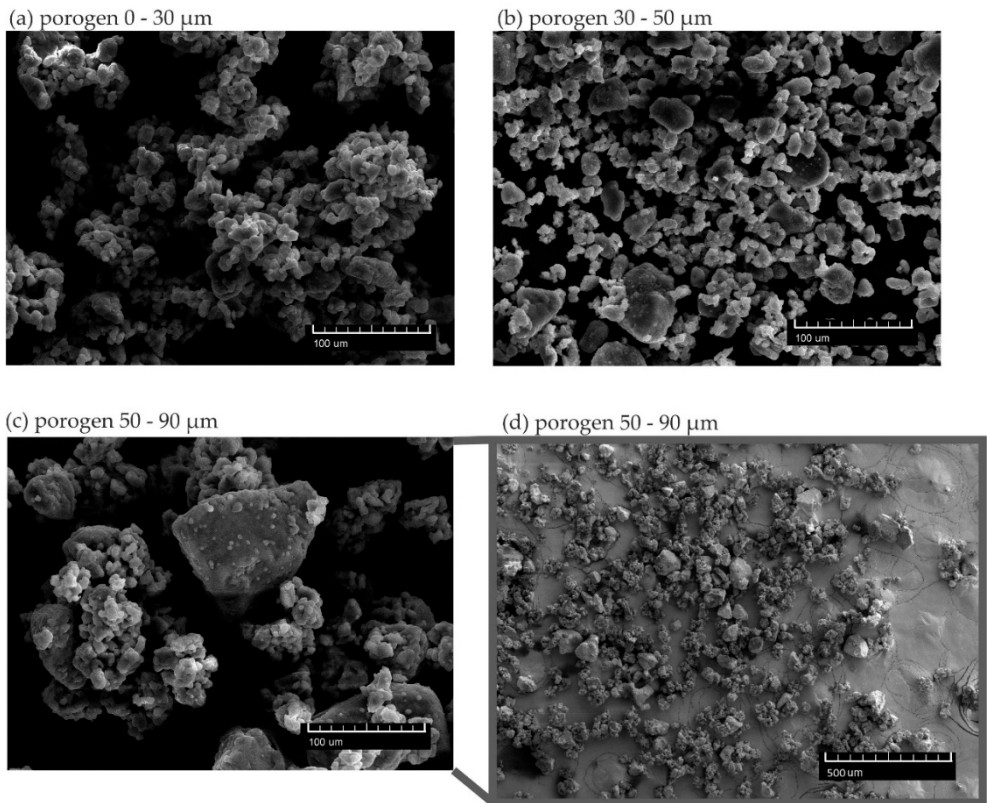

**Figure 3.** SEM microimages of NaCl particles that serve as a porogen: (**a**) 0–30, (**b**) 30–50, and (**c**) 50–90 μm. Image (**d**) is a lower magnification image of the 50–90 μm porogen.

Particles below the sieve lower size limit are present in all the salt fractions. In order to relate the porous morphology of gels with the porogen structure, a semi-qualitative evaluation of the porogen particles was performed. The size distribution for each batch was evaluated based on the SEM images of particles using the ImageJ software, with several tens of particles counted in each image of the three images for each size range. The porogen size distribution revealed a low number of two larger fractions in the ranges 0–30 and 30–50 μm, while the small particles 5–15 μm originating from clusters were always present, even in the batches sieved through higher sieve sizes; see Figure 4. The porogen fraction at 50–90 μm contained predominantly particles from 5 to 70 μm in size. The fractions above represent counts of a pore of respective size (black bars in Figure 4). However, when the distribution of sizes was recalculated to the volume distribution (grey bars in Figure 4), one can see that the most significant volume fraction in the system comprises particles of sizes corresponding to sieve ranges. The volumes were calculated for equivalent spheres of the diameter obtained from the SEM images (c.f., Table 3), with values of the most represented particle size volume-wise and the corresponding volume fractions. When incorporated into the gels and washed away, these particles should form the corresponding volume fraction of pores and pore channels.

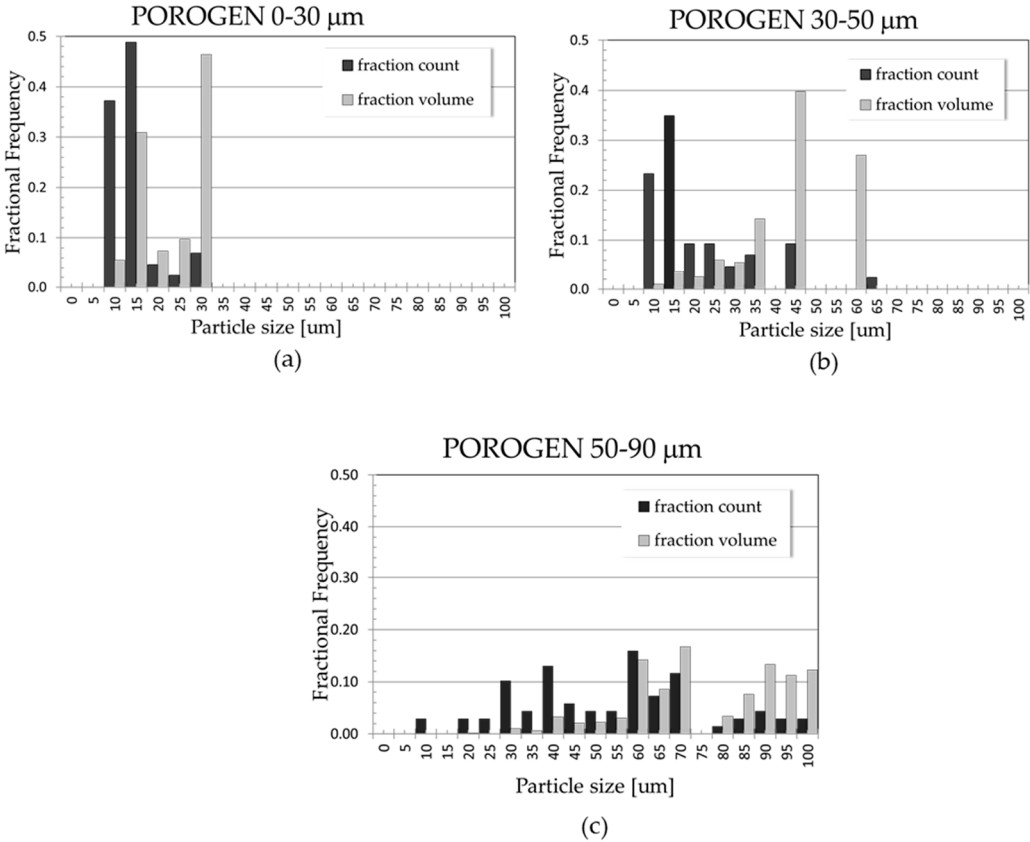

**Figure 4.** Porogen size distribution. Black columns—number distribution "fraction count"; grey columns—volume distribution obtained from SEM images using manual particle identification and counted using Image J.

**Table 3.** Porogen particle size distribution and its effect on the volume distribution. [a] Pore size as the diameter of an equivalent sphere. Determined from the SEM images of porogen particles; [b] volume of equivalent spheres of respective mean size; [c] the values for the two largest volume particle portions in each particle range are shown; maxima in bold.

| Porogen Particle Range (μm) | Mean Size of Particle Fraction [a] (μm) | Volume Fraction of Particles of Given Mean Size [b] (Pore vol. -%) |
|---|---|---|
| 0–30 | **30 ± 2.5** [c] | **46** |
|  | 15 ± 2.5 | 32 |
| 30–50 | **45 ± 2.5** | **40** |
|  | 35 ± 2.5 | 14 |
| 50–90 | **67 ± 5.5** | **40** |
|  | 92 ± 2.5 | 37 |

### 3.3. Pore Distribution Analysis

Morphological evaluation based on LSCM showed that the porogen particles can be correlated with the porogen assemblies imprinted in the scaffold matrices. Figure 5 shows the appropriate porogen size ranges and pore size distributions for the prepared samples analyzed by SEM and LSCM.

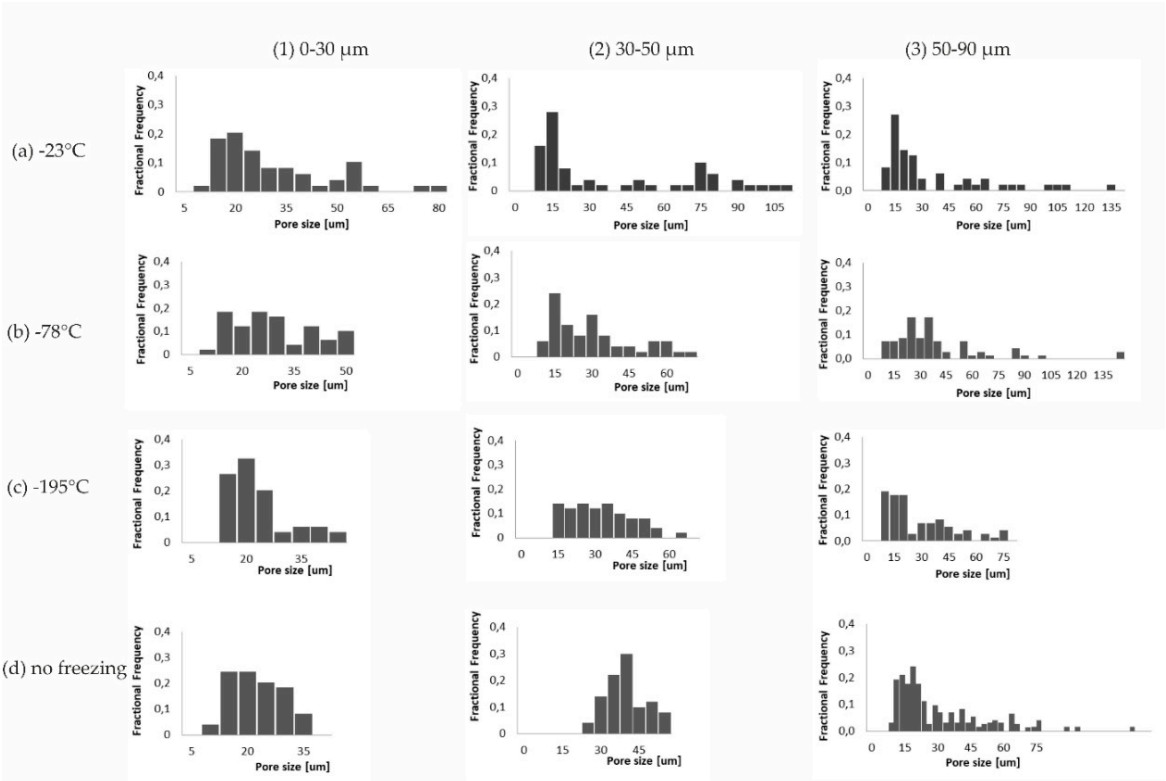

**Figure 5.** Fractional distribution of pore sizes in hydrogels. The pore size distribution suggests that the freezing method when applied before the morphological evaluation can influence the interpretation of the properties of the samples. It should be noted that the total pore number visually detected does not change depending on the freezing temperature, while the pore size as well as the gel wall thickness reveal some alterations. These changes in the pore sizes can be also tracked through the images (a–c) in Figure 1. There was significant development of large pores at the −23 °C used freezing temperature in all three groups of gels. The pore sizes in the gels prepared under milder freezing conditions at least doubled compared with those gels frozen at the −195 °C temperature or with the gels observed swollen in room temperature by LSCM. Pores between 15 and 25/30 μm represented a significant number of size fractions. This effect may be caused by the presence of "fine dust", which was discovered during the analysis of porogen particles harvested after fractionation. The distribution of pore sizes in the samples prepared using liquid nitrogen (195 °C) obtained by SEM can be compared to those images acquired by LSCM. Indeed, the difference between the porogen size distribution and the porous structure seen in the images from LSCM documents the differences caused by the change in gel due to the hydrogel swelling. The gel volume increases as reaches its equilibrium swelling state. This process starts with the leaching process, when substances from the preparation phase—e.g., the porogen and additives—are removed and the sample is washed. We assumed that the volume change upon swelling is isotropic, therefore the pores also expanded isotropically, as is shown in Figure 1. A certain similarity of structures obtained by LSCM and SEM with a freezing temperature of −195 °C was noticeable. However, the walls of the gels in the frozen samples were thinner compared with the walls of the gels in the equilibrium swollen state. The macroscopic volume of the samples was unchanged, which meant that the gel matrix volume change was compensated by the pore enlargement.

## 3.4. Swelling of Macroporous Hydrogels

The swelling of macromolecular gels depends on the gel chemical composition, network density, and interaction of the network with the swelling liquid, as well as the gel construct morphology. The equilibrium value of swelling provides an important structural parameter. Swollen macroporous hydrogels contain a significant amount of swelling liquid in their structure, located both in the voids and in the macromolecular network of the matrix. The total equilibrium amount of water in swollen

gels is then a sum of the two contributions, $V_{PL}$ and $V_{mx}$—c.f., volume balance and the derived relation expressed by Equations (2)–(7) in the experimental part. We have prepared a reference non-porous hydrogel of the same composition as the macroporous systems, except that no salt porogen was incorporated, and determined its equilibrium swelling. The swelling degree of neat pHPMA hydrogel, $sd_{mx}$, was 92 wt.-% (standard deviation was 1.6 wt.-%). Thus, the total porosity could be determined using the balance equations described in Section 2.9. Additionally, from the equilibrium weight (or volume) and the swelling of macroporous and non-porous hydrogels, the pore volume $\phi_{por}$ could be calculated. The resulting values of $\phi_{por}$ for the three macroporous systems were very close and revealed that around 60 % of the swollen macroporous hydrogels volume was occupied by pores—c.f., Table 4.

**Table 4.** Swelling of macroporous hydrogels determined gravimetrically; swelling estimated from computer reconstruction; volume fractions of porogen-formed pores calculated from swelling, $\phi_{PT}$, and large pore volume fraction; $\phi_{PL}$, estimated from reconstructed LSCM z-stacks (c.f. Section 3.5). [a] Swelling degree *SD* is expressed as the percentage of water in the equilibrium swollen gel according to Equation (1). *SD* value from the computer reconstruction was obtained by counting the matrix voxels and void voxels in the total depicted volume and from the experimentally determined swelling of the hydrogel matrix. The reference volume of pores estimated from the reaction mixture composition was 0.60 (volume of NaCl added). [b] $\phi_{PT}$ is the volume fraction of pores imprinted by salt particles calculated from equilibrium swelling. [c] $\phi_{PL}$ is the volume fraction of large pores according to computer volume reconstruction.

| Sets of Hydrogels | Swelling Degree $SD$ [wt.-%] [a] | | Pore Volume Fraction in Swollen Hydrogel | |
|---|---|---|---|---|
| | Gravimetric Experiment | Computed Reconstruction | $\Phi_{PT}$ [b] | $\Phi_{PL}$ [c] |
| pHPMA Matrix Neat | 92.0 ± 1.6 | NaN | NaN | NaN |
| 1 (0–30 μm) | 95.4 ± 1.5 | 94.6 | 0.62 | 0.32 |
| 2 (30–50 μm) | 95.7 ± 2.0 | 96.4 | 0.59 | 0.55 |
| 3 (50–90 μm) | 95.4 ± 1.8 | 94.3 | 0.59 | 0.29 |

Swollen macroporous pHPMA hydrogels contained in their structure significant portion of water, more than 95 %-wt; see the results of the gravimetric determination in Table 4. To compare with porosity calculation, the final pore volume was estimated from the known amount of salt particles used for the gel preparation; in our case, the volume of salt was approximately 60 % for each type of porogen. The particle filling corresponded well to the determined porosity from the swelling balance—c.f., $\phi_{PT}$ values in Table 4. We noticed that the gels prepared with PEG after washing exerted opacity, which could be caused by some light scattering heterogeneity present in the swollen gel. We assumed that PEG caused the phase separation of the forming pHPMA chains; therefore, after the washing of the gels, the formed hydrogel macromolecular network shrank and its final volume should be corrected for the removed amount of PEG. Including such a correction, the total pore volume in the swollen hydrogels would increase to approx. 80%, but it is difficult to prove that number by light microscopy imaging methods, as the secondary pores were of a size below the light microscopy resolution limit. In addition, we computed the fraction of "pore" voxels from the 3D computer reconstructions based on LSCM z-stacks, and used the experimental correction of the neat matrix to determine the total swelling from the combined LSCM z-stacks (c.f., Table 4). The swelling values determined from the gravimetric experiment and obtained from the image analysis of the LSCM z-stacks of swollen gels were in a good agreement, as shown in Table 4.

Finally, the equilibrium swelling degree of hydrogels determined after the freezing and drying treatment was 95 ± 0.7 wt.-%, which was in a good agreement with the values determined with primary gels. This agreement suggests that the freezing and drying steps did not cause any significant internal failure of hydrogels, which would increase the swelling or lead to sample disintegration caused by

re-swelling. The swelling value is related to the gel structure; it provides an estimate of the total pore volume of porogen particles. Moreover, swelling can serve as check of the attempt to quantify the computer reconstructed images. In the case of pHPMA macroporous gels, this comparison suggests that the computer processing of images will need further mathematical treatment and possibly a finer method of fluorescent dye incorporation to obtain micrographs with a better contrast. Considering the high complexity of the studied structures, the incipient results of the 3D reconstruction quantification are promising and provide direction for future research.

### 3.5. 3D Reconstruction of the Swollen Hydrogel Volume

LSCM allowed a more in-depth analysis and provided z-plane scans within a range of several tens of micrometres (30, 50, and 90), resulting in a series of z-stacks for each material. The hydrogel volumes were reconstructed from z-stacks using the Imaris imaging software package (see Figure 6).

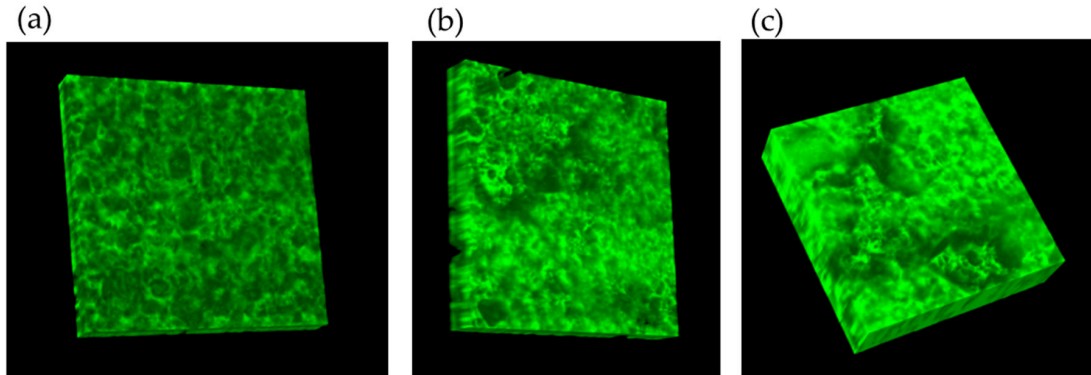

**Figure 6.** Examples of hydrogel structures from the LSCM z-stacks reconstructed by 3D computer image analyses using the Imaris software package and imaged using FIJI, an Image J Volume viewer plug-in. The x and y lengths of the scanned volumes correspond to 400 μm. (**a**) Salt size 0–30 μm, (**b**) salt size 30–50, (**c**) salt size 50–90 μm.

The software allows for quantification and further mathematical image exploration. We evaluated first the portion of pores made by porogen imprints (Table 4); the fine pores formed by phase separation in gel matrices and partially by fine salt particles were not visible in light microscopy. The computer image processing software can provide parameters such as the pore size distribution, volume of connected space, pore surface area, etc. The fraction of large porogen-formed pores was evaluated and used to calculate swelling (Table 4). Examples of the three reconstructed hydrogel slabs are shown in Figure 6, revealing the limitations of light microscopy when trying to scan deeper layers of samples; the images show the largest possible z-axis ranges achieved with these samples, while the scanning of deeper z-stacks was not technically possible due to the limited material transparency. The mechanical cutting of the hydrogel slice from the gel surface fixed in the microscope and the continuation of z-scanning could be a future solution. In the case of particles larger that the scanned gel volume, the image overemphasizes porosity. However, when a larger area of gel is scanned, the effect of exceeding the pore volume is diminished.

### 3.6. Cell Cultivation on Hydrogels

Nowadays, the properties of hydogels for cell growth are widely studied [37]. In our study, the prepared hydrogels served as a model for demonstrating rMSC growth and hydrogel morphology, which can be observed by LSCM at the same time. The aim was to determine which pore size was most appropriate for rMSC growth. The cell cultivation study was performed by applying rMSC to sample sets of hydrogels (1, 2, 3) at an initial seeding cell concentration of $1 \times 10^5$ cell scaffolds, conducive to growing cells. The cell growth within the hydrogels was also visualized with a confocal

microscope using Hoechst fluorescent labeling. LSCM allows observation in real time, and examples of hydrogel with labeled cells (nuclei visible as blue spots) are shown in Figure 7a. After five days of cultivation, the cells were distributed throughout the hydrogels, regardless of pore size, with a density of $3 \times 10^4$ cells. The cells proliferated well, although the initial cell attachment was low (Figure 7b).

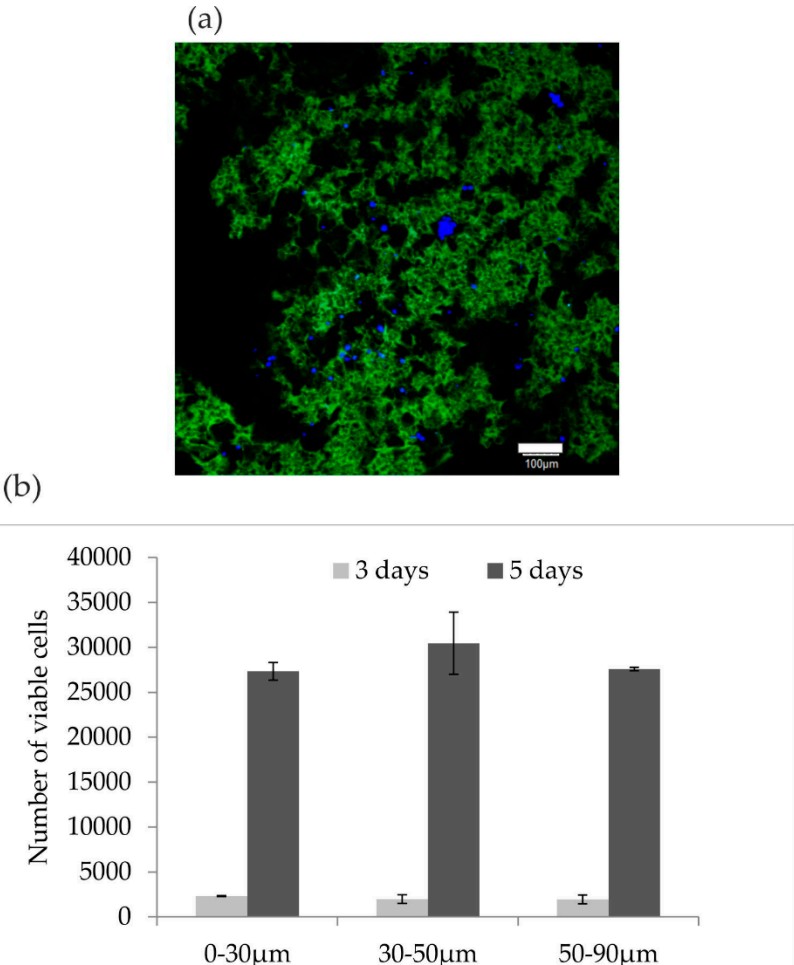

**Figure 7.** (**a**) rMSC growth on the HPMA-based hydrogel scaffold with porogen 30–50 μm was determined by LSCM after 5 days and visualized using cell nuclei fluorescence (blue spots). Green structures are the gel scaffold fluorescence visualized due to the presence of fluorescein in the polymer structure. (**b**) The average number of viable cells growing on the gel scaffolds at day 3 and 5. The amount of initially plated cells was $1 \times 10^5$ per scaffold sample. The growth of rMSC in larger pores is documented by a movie from the LSCM images in ESI. LSCM allows one to focus on levels below the material surface; in this case, cells as deep as 100 μm could be observed within the scaffold bulk.

## 4. Conclusions

Three types of chemically identical soft porous hydrogels scaffolds based on pHPMA were prepared. The macroporous structure of the hydrogels was created using NaCl particles as porogen that imprint the porous structure into the scaffolds. The porogen particles were fractionated to fine size fractions of 0–30, 30–50, and 50–90 μm. The internal morphology of the macroporous hydrogels was investigated by SEM and LCSM. The freeze-drying methods employed for SEM analysis caused severe changes in the porous structure of the hydrogel; LSCM imaging allowed the visualization of the relevant spatial arrangement within the hydrogels in their native swollen state. Nonetheless, SEM is

a very useful technique for the visualization of morphological details in the micron range, offering sharp focus and a large depth of field. However, great care must be taken in the sample preparation steps, as the sub-zero temperature treatment can cause changes in soft hydrogels. We found that the LSCM can reveal a more realistic morphological structure of swollen hydrogels in comparison with SEM. Moreover, LSCM is a relatively quick method and allows the accurate assessment of the size and distribution of the pores in soft swollen macroporous hydrogels. The thickness of the walls between the pores was revealed to be thicker compared to the SEM images due to the fact that the hydrogels were scanned in an undisturbed swollen state. Furthermore, LSCM revealed the detailed porogen size distribution and presence of fine particles below the sieve size limit stemming from the preparation method. The swelling values of the reference non-porous pHPMA matrix and macroporous hydrogels provided values of porosity that was around 60 % and corresponded well to the porosity estimated from reconstructed images. It is anticipated that LSCM will allow an analysis of the inner structure of the hydrogels in their native state, showing real-time regions of seeded cells with respect to their true morphology. The LSCM method can help us minimize the mistakes connected to the morphology determination of hydrogel scaffolds which can later influence the application of these materials for tissue engineering purposes.

**Supplementary Materials:** The following are available online at http://www.mdpi.com/2076-3417/10/19/6672/s1: Figure S1: Examples of the evaluation of size of pores determined using the ImageJ software.

**Author Contributions:** The individual contributions of the authors: Conceptualization, B.P., O.J., M.D.-S.; methodology, E.C.-K., B.P., M.V., and L.K.; validation, N.R.B., B.P.; formal analysis, B.P., M.Š.; investigation, B.P., M.V., E.C.-K.; resources, B.P., M.V.; data curation, M.Š., M.D.-S., O.J.; writing—original draft preparation, B.P., O.J., M.D.-S.; writing—review and editing, B.P., O.J., M.D.-S., and M.V.; visualization, N.R.B., B.P., M.V.; supervision, O.J. All authors have read and agreed to the published version of the manuscript.

**Funding:** The authors gratefully acknowledge financial support by the Czech Science Foundation, Project No. 17-11140S, No.17-08531S, Czech Academy of Science (# MSMJ200501801).

**Conflicts of Interest:** The authors declare no conflict of interest.

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
