# Peer review of "Revealing the True Morphological Structure of Macroporous Soft Hydrogels for Tissue Engineering"

_applsci, doi:10.3390/app10196672_

Round 1

Reviewer 1 Report

Remarks

  1. „Revealing the True Morphological Structure of Macroporous Soft Hydrogels for Tissue Engineering”

What does “True Morphological Structure” mean? Please explain it.

  1. “The SEM was performed with specimens rapidly frozen to various temperatures, while non-frozen gels were visualized with LSCM. (3&4) Results & Conclusion: In comparison to LSCM, the SEM images revealed significant alteration of mean pore size and appearance of newly formed multiple connections between the pores depending on the freezing conditions”.

If the freezing process affects the morphology and structure of the samples, then is it correct to compare the SEM images obtained for samples rapidly frozen at different temperatures with the LSCM images obtained for non-frozen samples?

  1. LSCM visualization aided understanding of the dynamics of pore generation using sodium chloride, providing direct observation of hydrogel scaffolds with the growing cells”.

In this case the LSCM visualization of the dynamics of pore generation (changes in time) should be added.

Author Response

  1. „Revealing the True Morphological Structure of Macroporous Soft Hydrogels for Tissue Engineering”

What does “True Morphological Structure” mean? Please explain it.

Answer:  We try to point out the problems which can happen during characterization of morphology of hydrogels. During the characterization using SEM, the morphology can be dramatically changed, which was shown in our manuscript. We tried to use method of visualization of morphology, which can´t interfere with the swollen state of the hydrogels. From this description, we consider the scaffold in its natural swollen state as the structure with “true morphology”. In our case, these are hydrogels in the native hydrated state. The aim was to demonstrate the morphological structure of the hydrogel by undistorted sample preparation for hydrogel visualization.

  1. “The SEM was performed with specimens rapidly frozen to various temperatures, while non-frozen gels were visualized with LSCM. (3&4) Results & Conclusion: In comparison to LSCM, the SEM images revealed significant alteration of mean pore size and appearance of newly formed multiple connections between the pores depending on the freezing conditions”.

If the freezing process affects the morphology and structure of the samples, then is it correct to compare the SEM images obtained for samples rapidly frozen at different temperatures with the LSCM images obtained for non-frozen samples?

 Answer:  The SEM methods is a conventional method for visualization the morphology of hydrogels. This study revealed the weaknesses of the SEM method compared to the LSCM method. The main advantage of the LSCM method is the fact that the samples do not have to be frozen before visualization and the LSCM images provide a "true morphological structure".

  1. LSCM visualization aided understanding of the dynamics of pore generation using sodium chloride, providing direct observation of hydrogel scaffolds with the growing cells”.

In this case the LSCM visualization of the dynamics of pore generation (changes in time) should be added.

Answer:  The hydrogel pores were formed with sodium chloride before LSCM visualization. The prepared hydrogels were not degradable. The hydrogels in their swollen state are stable without changes of their morphology. Moreover for our study, we used short term incubation with rMSC which can not  changed the morphology of this types of nondegradable pHPMA gels.

Reviewer 2 Report

This work is interesting while it needs some minor revisions;

1. pls provide rationale why the pore range is chosen in this study.

2. key recent works have shown the physical roles of hydrogels that can encapsulate stem cells. Thus the authors should reference these and discuss properly to feedback the recent research trend in this field as referenced below:

- Chaudhuri, et al., Hydrogels with tunable stress relaxation regulate stem cell fate and activity, Nature materials 15(3) (2016) 326-334

- Lee et al. Emerging properties of hydrogels in tissue engineering. J Tissue Eng, 2018; 9: 2041731418768285.

3. cell morphology on different scaffolds also needed.

4. the big difference in cell number between day 3 and 5 is strange; authors need to show with other method like cell viability assay.

Author Response

  1. pls provide rationale why the pore range is chosen in this study.

Answer:  The range of pores 0-30 μm, 30-50 μm and 50-90 μm was chosen to evaluate how the freezing can changed the structure of hydrogels of different porosity. These findings could be important in case of characterization of prepared hydrogels with different porosity for various tissue engineering purposes.

  1. key recent works have shown the physical roles of hydrogels that can encapsulate stem cells. Thus the authors should reference these and discuss properly to feedback the recent research trend in this field as referenced below:

- Chaudhuri, et al., Hydrogels with tunable stress relaxation regulate stem cell fate and activity, Nature materials 15(3) (2016) 326-334

- Lee et al. Emerging properties of hydrogels in tissue engineering. J Tissue Eng, 2018; 9: 2041731418768285.

Answer:  We thank, for this comment. We add the second recommended citation to our manuscript. Our study was closely focused to the comparison of methods which can be used for evaluation of hydrogel morphology.  Moreover we used the cell growth on our hydrogels as a example of advantage of LSCM in visualization hydrogels together with cells in one.  The cells were seeded on the hydrogels after their preparation, so the cells are not affected by gelation and processes coupled with formation of gels. Although the direct interaction of stem cells with hydrogels during gelation is a very interesting and important area, especially in field of regeneration of bones or cartilages, we did not used this system in our studies.

  1. cell morphology on different scaffolds also needed.

Answer:  Rat MSC were used only as a model to demonstrate the susceptibility of pHPMA for cell growth and moreover for the „in situ“ visualization of cells together with morphology of hydrogels. Our study was not focused to the detailed evaluation of cell adhesion/growth/proliferation in dependence of the type of hydrogels.

  1. the big difference in cell number between day 3 and 5 is strange; authors need to show with other method like cell viability assay.

Answer:  Thank you for this question. We used for the evaluation the amount of cells, cell viability test – AlamarBlueTM –cell viability reagent (Thermo Fischer Scientific), which is based to the compound resazurin, which is metabolized in viable cells to fluorescently active resorufin. The amount of cells was calculated from calibration curve and all of them were methabolicly active, based on fluorescence of resorufin.  Evaluation of cell growth was used  as a example of possibility to monitore morphology of hydrogels together with cell growth. We did not concentrate to the detailed study of cell growth and proliferation as in other our studies e.g. (Pradný M., Journal of Polymer Research 21(11),(2014); Janouskova O.,Biomedical MAterials 14(5), (2019))